# Historical nitrogen fertilizers use in China from 1952 to 2018

Zhen Yu[1,2], Jing Liu[2], Giri Raj Kattel[3,4,5]

[1] Collaborative Innovation Center on Forecast and Evaluation of Meteorological Disasters (CIC-FEMD), Nanjing University of Information Science & Technology, Nanjing, China 210044

[2] Institute of Ecology, School of Applied Meteorology, Nanjing University of Information Science and Technology, Nanjing, 210044, China

[3] School of Geographical Sciences, Nanjing University of Information Science & Technology, Nanjing, China

[4] Department of Infrastructure Engineering, University of Melbourne, Parkville, Melbourne, Australia

[5] Department of Hydraulic Engineering, Tsinghua University, Beijing, China

*Correspondence to*: Zhen Yu (zyu@nuist.edu.cn)

**Abstract.** China ranks the highest position on the nitrogen (N) fertilizers consumption in the world. Although the N fertilizers use has greatly contributed to the China's food production, this has also caused unprecedented alteration in the biogeochemical
cycles and endangered terrestrial and aquatic ecosystems. Existing use of N fertilizers in China as shown by digital maps are usually coarse in resolution, and intermittently covered with biasedly gridded dataset. Here, we have reconstructed a historical, annual N fertilizers use dataset in China and resampled it to 5 km × 5 km resolution covering the period of 1952 to 2018 by integrating improved cropland maps. Results showed that the most of the N input was directly applied as N-only fertilizer, while the contribution from compound fertilizers ranged between 16% and 24% since 1980. The national total N fertilizers
input increased from 0.06 Tg N yr$^{-1}$ (0.05 g N m$^{-2}$ yr$^{-1}$) in 1952 to 31.15 Tg N yr$^{-1}$ (18.83 g N m$^{-2}$ yr$^{-1}$) in 2014, then decreased to 28.31 Tg N yr$^{-1}$ (17.06 g N m$^{-2}$ yr$^{-1}$) in 2018. Despite the total N input decreased by 9.1% (2.84 Tg N yr$^{-1}$) from 2014 to 2018, the N input from compound fertilizers increased by 6% (0.43 Tg N yr$^{-1}$) during the corresponding period. The previous FAO-data-based N fertilizer products in China overestimated the N use in low, but underestimated in high cropland coverage areas. However, our newly reconstructed data have not only corrected the existing biases and improved the spatial distribution
but also showed vegetable and other crops (orchards), but not grain crops, are the most intensively fertilized crops in China, implying the importance of quantifying greenhouse gas (GHG) emissions from these croplands. We argue that the reconstructed, spatially-explicit N fertilizers use data in this study are expected to contribute to better understanding in biogeochemical cycles including the simulations of GHG emission and food production in China. The spatial-explicit N fertilizer use and the crop-specific N fertilizer use datasets is available via an open-data repository
(https://doi.org/10.6084/m9.figshare.21371469.v1) (Yu, 2022).

# 1 Introduction

The birth of the Haber-Bosch technique has converted enormous amount of unreactive nitrogen (N) to reactive forms greatly alleviating N limitation in the agricultural production (Galloway et al., 2004; Sutton and Bleeker, 2013; Erisman et al., 2008; Lu and Tian, 2017). However, the excessive N fertilizers use in crops has become a global issue as this has caused unprecedented alteration of terrestrial and aquatic ecosystems. Since the introduction of synthetic N fertilizers in China in the early 1910s, its use has markedly increased along with population growth and agricultural intensifications. Today, China ranks highest in producing and using the reactive N for food and fiber production (Naughton, 2006; FAOSTAT database, 2018). About 30% of global N fertilizers was applied in China's cropland in 2017 alone, which accounted for about 9% of the global cropland areas  (FAOSTAT database, 2018; Yu et al., 2021). Despite such high intensity of N fertilizers use, the crop yields in China were lower compared to the global average (Wang et al., 2020b; FAOSTAT database, 2018). Former study reported an increase of N fertilizers use by over 2.7 folds from 1977 to 2005, while the marginal contribution of N fertilizers to food production was declining, e.g. the large increment of N fertilizers use helped enhance grain yields by only 98% (Ju et al., 2009). Both field-based and modeling studies evidenced the widespread, over-use of N fertilizers in China, and it has been advocated that reducing N fertilizers use would not be adversely affecting the crop production (Zhu and Chen, 2002; Ju et al., 2009; Tian et al., 2012; Huang et al., 2008). Reduced N fertilizers use has implications for minimal food-production costs and higher environmental benefits including the reduced water-borne pollution and greenhouse gas (GHG) emissions  (Kahrl et al., 2010).

One of the direct benefits of reducing the N fertilizers use in cropland has been reported as the reduction in the $N_2O$ emission – a GHG with  global warming potential as high as 298 times  greater than that of $CO_2$ over a 100-year time horizon (Myhre et al., 2013). China is currently the largest $N_2O$ emitters worldwide, in which agriculture alone has contributed to 64% (Shang et al., 2019; Zhou et al., 2014), implying the potential of  $N_2O$ emission reduction via optimization of N fertilizers use in cropland (Tian et al., 2012). Moreover, the reduced N fertilizers use in cropland also helps alleviate soil acidification and eutrophication further benefitting the terrestrial and aquatic environments (Strokal et al., 2016). A better understanding of the spatial-temporal distribution of N fertilizers use in China's cropland will help locate the hot-spot of N surplus and eventually contribute to the management including the efficient use and optimization. However, N fertilizers use in China has greatly varied spatially and temporally with disparity in the dataset to capture the historical N fertilizers use. Most existing spatio-temporal dataset depicting N fertilizers use in croplands of China are irregular and relatively coarse. For example, the N fertilizers use dataset are available in both international and national archives. The International Fertilizer Industry Association (IFA) and the Food and Agricultural Organization (FAO) which have archived the amount of annual N fertilizers consumption at national level since 1961, while the National Statistical Bureau of China has archived the amount of annual N fertilizers use at both national and provincial levels since 1987. Some attempts were made to spatialize the national level N fertilizers use, but the products were still relatively low in spatial resolution (e.g. 0.5 degree for Lu and Tian (2017)). Many process-based modeling studies simulated the impacts of N fertilizers use on biogeochemical cycles in China using the low-resolution N fertilizers dataset, which greatly impaired the reliability of the estimations (Tian et al., 2012; Wang et al., 2020a). There is an

urgent need for a long-term, spatial explicit N fertilizers dataset to serve the quantification of national and global GHG budgets, and to benefit data analyses and the environment protection including reduced water pollutions and improved land-based ecosystem functions.

In this study, we have reconstructed the annual N fertilizers use in China's cropland using various statistical records, reports, and gridded images at 5 km × 5 km resolution covering the period of 1952 to 2018. We aimed to: 1) develop a continuous dataset depicting the N fertilizers use in cropland in China; and 2) examine the historical distributions and shifts of N fertilizers use cropland in China. Our focus is on the chemical, rather than the organic N fertilizers use, and the term 'N fertilizers' we have adopted here is to refer exclusively to synthetic N fertilizers.

## 2 Data and methods

### 2.1 Reconstruction of the national and provincial N fertilizers use

This study focuses on N fertilizers use in the mainland China, while Taiwan, Hong Kong, and Macaw were excluded due to no data availability. The dataset we have adopted here for synthetic N fertilizers use in cropland of China was from N-only fertilizers, and N-mixed fertilizers or 'compound fertilizers' (see equation 1).

$$N_{tot} = N_{only} + N_{mix} \qquad (1)$$

where $N_{tot}$ indicates the total N use in China's cropland, $N_{only}$ indicates the N fertilizer use from N-only fertilizer, and $N_{mix}$ indicates N-mixed fertilizer use. We have adopted two different phrase and units to differentiate the N fertilizers use at national level i.e. 'N fertilizers input' (Tg N per year), and crop field level i.e. 'N fertilizers use rate' (g N per unit land per year).

We first reconstructed the total N fertilizers input in China's cropland at national level. The national fertilizer inputs were provided by both FAO (https://www.fao.org/faostat) and the National Bureau of Statistics of China (Chinese Statistical Yearbook (CSY); also available from https://data.stats.gov.cn). Specifically, the FAO data provides the total N input in China from 1960 to 2018. While in comparison, the CSY data describes total fertilizers use (including N, phosphate, potassium fertilizers such as ammonium phosphate) covering the period of 1952 to 2018, and the N-only fertilizers (e.g. ammonia, ammonium nitrate) from 1987 to 2018. The national total N fertilizer input of the period 1960 to 2018 was directly obtained from the FAO, while the ratio of N fertilizers to the total fertilizers was used to derive the total N fertilizers use from 1952 to 1959.

Second, we compiled the N fertilizers used in each province in China. For the period 1987-2018, N fertilizers use was directly derived from the CSY database. For the period 1952 to 1986 when the provincial data was unavailable, the reconstructed national N fertilizers use was allocated to each province based on the provincial N proportions derived in 1987.

**2.2 Reconstruction of the crop-specific N fertilizers use rate in each province**

We examined the major crops planted in China and grouped them into 10 types, including early rice, mid-season rice, late rice, wheat, corn, soybean, oil seeds, cotton, vegetable, and other crops. Specifically, other crops include barley, sorghum, sugarcane, tobacco, fruits (e.g. apple, pear, citrus). The N fertilizers use rate for each major crop types (except other crops) was intermittently reported in the Cost-benefit Report of the National Agricultural Products (CBR) covering the period of 2004-2018 (Table 1) (CBR data can be obtained from: https://data.cnki.net/trade/Yearbook/Single/N2021120200?zcode=Z009). The CBR data provides officially released fertilizers use information summarized from thousands of samples collected in each province in China. First, we created an empty table to record the N fertilizer use rate for each province with all the 10 types included. Second, the N use rate of the table was allocated using data obtained from CBR when available. Third, if a crop type was never planted in the province, the N use rate was set to 0. Fourth, we checked and gap-filled the missing N fertilizers use rates in the province. For crop type with N use rate intermittently reported, we linearly interpolated the rate using the two nearest data reported before and after the year (see equation 2). While for crop type been planted in the province but its N fertilizer use were never reported, two fertilizers use scenarios were considered. For the first scenario, we assumed that the N fertilizers use rate of the crop in the province was the same as the average rate at national level (see equation 3). While for the second scenario, we assumed that the N fertilizers use rate of the crop in the province was the average of the rates adopted in nearby provinces (see equation 4).

$$Rate_i = (Rate_j - Rate_k) \times \frac{i-j}{k-i} \qquad (2)$$

$$Rate_{p,i} = \left(\sum_{q=1}^{n} Rate_{q,i}\right)/n \qquad (3)$$

$$Rate_{p,i} = \left(\sum_{q=1}^{m} Rate_{q,i}\right)/m \qquad (4)$$

where $Rate_i$ indicate the N fertilizer use rate of a crop type in year $i$, while $j$ and $k$ indicate the nearest data reported in years $j$ and $k$ representing the years before and after the year $i$. $Rate_{p,i}$ indicate the N fertilizer use rate of a crop type in province $p$ in year $i$, $q$ indicates province $q$ with the rate available in CBR, $n$ indicate total number of province with N fertilizer rates available in CBR, and $m$ indicate the total number of nearby province with N fertilizer rates available in CBR.

For the period of 1981 to 2004, the N fertilizers use rates were calculated from the N inputs and the planted areas of each crop type in each province. While for the period of 1952 to 1980 when provincial, crop type-specific N fertilizers use data were unavailable, we proportionally adjusted the N fertilizers use rate of each crop type based on the ratio of N fertilizers use in the year and the amount used in 1981 (see equation 5). This is assuming that the change of N fertilizers use rate of a crop is proportional to the change of the total N fertilizers use in the province.

$$Rate_{p,i} = Rate_{p,1981} \times N_{tot,p,i}/N_{tot,p,1981} \qquad (5)$$

where $Rate_{p,1981}$ indicate the N fertilizer use rate of a crop type in province $p$ in 1981, $N_{tot,p,i}$ indicate the total N fertilizer input for province $p$ in in year $i$, and $N_{tot,p,1981}$ indicate the total N fertilizer input for province $p$ in in year 1981.

As pointed out in the section 2.1, the total N input includes two components, i.e. N from compound fertilizer and N-only fertilizer. Unfortunately, CSY database documented the use of compound fertilizers in major crop types but the N ratio was not specified. Since the N ratio varied between 16%-33.33% in compound fertilizers according to the major fertilizers used in China (http://fgw.kaifeng.gov.cn/info/2107), we assumed two extreme scenarios that the N ratio were either 16% or 33.33% for compound fertilizer being applied to each major crops.

In addition, N fertilizers use in vegetables were also highly uncertain. Specifically, various vegetables were planted in China, while the N fertilizers use rates were missing for the most of the vegetables. However, the vegetables received much higher N fertilizers (71.9 g N m$^{-2}$) than non-vegetable crops according to the former published study (Huang et al., 2017). In general, the total fertilizers applied in vegetable was about 3.3 times higher than the recommended application rate (Huang et al., 2017). Therefore, two additional scenarios were considered in reconstruction of the N fertilizers use in vegetables. In the first scenario, we assumed that the N fertilizers use rate in vegetable was the same as the average of all other major crop types. In the second scenario, we assumed the N fertilizers use rate in vegetable was 3.3 times of the average rate of other major crop types. Therefore, we considered three uncertainty sources in this study (Table 2).

After the gap-filling of the nine of the ten major crop types (except other crops), the total N inputs were calculated by multiplying the rates and the areas of each crop type in each province. The provincial residue N inputs (the difference of the total N inputs calculated, and the total N inputs derived from the FAO and the CSY) were allocated to other crops and the N fertilizer rates were calculated by dividing the residue N inputs to the planting areas obtained from the China Agricultural Yearbook (Table 1).

$$Other\_Rate_{p,i} = [N_{tot,i} - \sum_{t=1}^{9}\sum_{q=1}^{n}(Rate_{q,i,t} \times Area_{q,i,t})]/Area_{p,i} \qquad (6)$$

where $Other\_Rate_{p,i}$ indicate the N fertilizer use rate of other crops in province $p$ in year $i$, $N_{tot,i}$ indicates the total N input in year $i$, $Rate_{q,i,t}$ indicate the N fertilizer use rate of crop type $t$ in province $q$ in year $i$, $Area_{q,i,t}$ indicate the planted area of crop type $t$ in province $q$ in year $i$, and $Area_{p,i}$ indicate the planted area of other crops in province $p$ in in year $i$.

Table 1. Datasets used for nitrogen (N) fertilizers use reconstruction.

| Datasets | Year | Resolution | Variable | Sources |
|---|---|---|---|---|
| Cropland distribution maps | 1900–2016 | Annual, 100m-5km | Cropland distribution | (Yu et al., 2021) |
| China Agricultural Yearbook (CAY) | 1980–2018 | Annual, provincial | Planted areas of each major crops in each province | National Bureau of Statistics of China |
| China Statistical Yearbook (CSY) | 1952–2018 1987–2018 | Annual, provincial | Total fertilizer N fertilizer | National Bureau of Statistics of China |

| | | | | |
|---|---|---|---|---|
| Cost-benefit Report (CBR) | 2004-2018 | Intermittently, provincial | N fertilizers use by crop types in each province | National Development and Reform Commission of China (Price Department) |
| FAO N fertilizer | 1960-2018 | Annual, national | Total nitrogen fertilizers use in China | (FAOSTAT database, 2018) |
| Rotation maps | 1980, 1990, 2000, 2002, 2011 | County-level | Crop rotation information | (Liu et al., 2018) |

Table 2 The eight scenarios considered in reconstructing nitrogen fertilizer use in China

| | Uncertainty sources | Scenarios | | | | | | | |
|---|---|---|---|---|---|---|---|---|---|
| | | 1 | 2 | 3 | 4 | 5 | 6 | 7 | 8 |
| 1 | Gap-filling approach | Nation[*] | Nation | Nation | Nation | Nearby[**] | Nearby | Nearby | Nearby |
| 2 | N ratio in compound fertilizer | 16% | 16% | 33.33% | 33.33% | 16% | 16% | 33.33% | 33.33% |
| 3 | Vegetable N fertilizer use | Ave[***] | Ave3.3[****] | Ave | Ave3.3 | Ave | Ave3.3 | Ave | Ave3.3 |

[*]: the N fertilizer rate of missing crop type(s) was derived from national average; [**]: the N fertilizer rate of missing crop type(s) was derived from nearby province; [***]: the vegetable N fertilizer use rate was assumed to be the average of other nine major crops; [****]: the vegetable N fertilizer use rate was assumed to be 3.3 time of the average rate of other nine major crops.

## 2.3 Approach for spatializing N fertilizers use

Before the N fertilizers use rate could be allocated spatially, a crop type map is required. Here, we reconstructed crop rotation maps from 1952 to 2018 using the model we previously developed (Yu and Lu, 2018; Yu et al., 2019) (see Figure S1 for the details). Reconstruction of annual crop rotation map can be divided into two periods, namely the periods before and after 1980. For the period of 1980-2018, county-level crop rotation maps in 1980, 1990, 2000, 2002, and 2011 were used (Table 1) (Liu et al., 2018). More specifically, when allocating a crop type in a province for a year from 1980, the cropland grid-cell located in a county was given priority to be assigned to the crop type identified from the nearest-year rotation map. Due to the lack of data, the crop rotation map in 1980 was used for the period before 1980.

Based on the 100-m crop rotation maps developed in the previous step, we link the crop type and the N fertilizers use rate developed for each crop types in each province. Specifically, we spatialized the N fertilizers use rate for each year using the 100-m crop type maps. For grids with multiple crops cultivated in a year, the fertilizers use rate was the total N fertilizers use of all crops (i.e., total N fertilizer applied in a grid-cell in a year). The 100-m resolution N fertilizers use rate maps were then resampled to 5 km × 5 km for a comparison and analyses. In this study, we considered three uncertainty sources and the uncertainties of the N fertilizer maps were derived from the eight scenarios listed in Table 2. The methodology flowchart is showed in Figure 1.

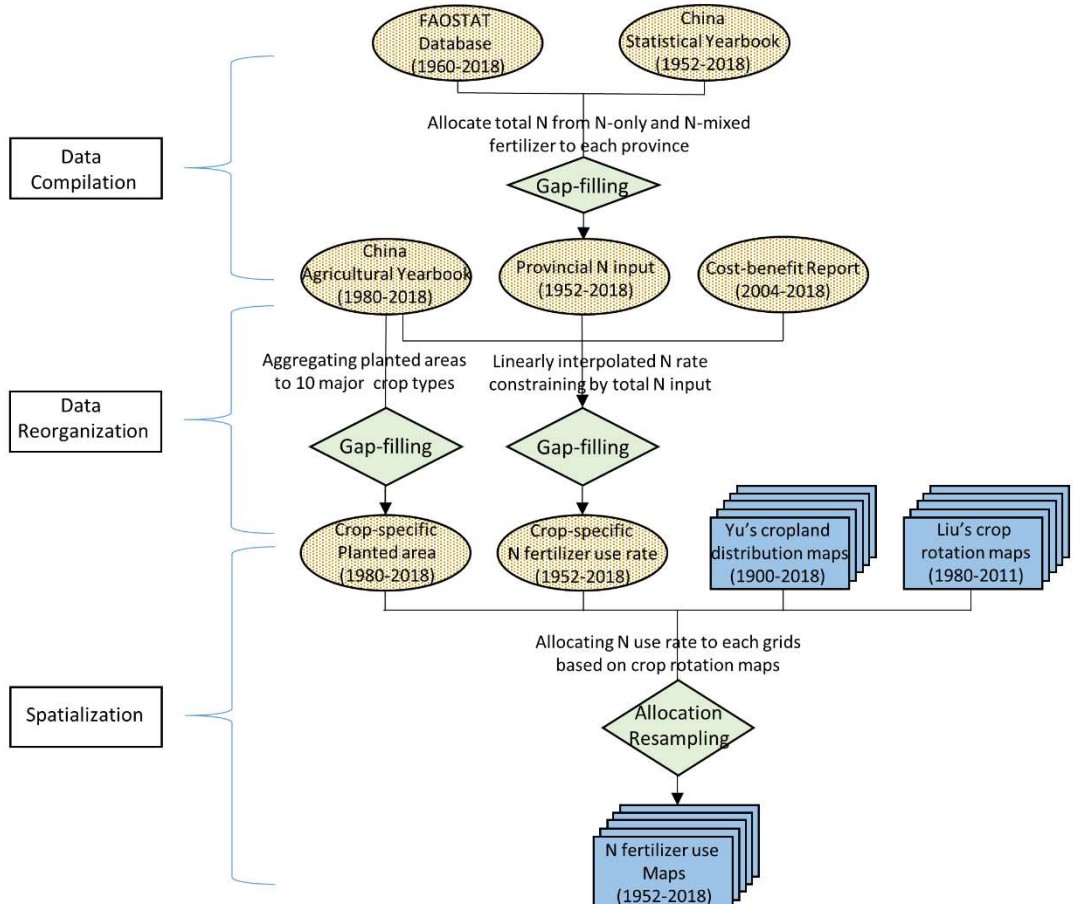

Figure 1. Methodology flowchart of nitrogen (N) fertilizer map reconstruction

## 3 Results

### 3.1 National nitrogen fertilizers use for crop production in China

The compiled national total N fertilizers use increased from 0.06 Tg N yr$^{-1}$ in 1952 to the peak of 31.15 Tg N yr$^{-1}$ in 2014 then decreased to 28.31 Tg N yr$^{-1}$ in 2018. The majority of the N input was directly applied as N-only fertilizers (e.g. urea, ammonium carbonate), while the contribution from compound fertilizers (e.g. ammonium phosphate) increased from 16% in 1980 to 24% in the 2010s. Despite the total N input decreased by 9.1% (2.84 Tg N yr$^{-1}$) from 2014 to 2018, the N input from compound fertilizers increased by 6% (0.43 Tg N yr$^{-1}$) during the corresponding period (Figure 2).

The rising N fertilizer input was consistent with the N fertilizers use rates (per square meter of cropland), which increased from 0.05 g N m$^{-2}$ yr$^{-1}$ in 1952 to 18.83 g N m$^{-2}$ yr$^{-1}$ in 2014, then decreased by 9.4% from 2014 to 17.06 g N m$^{-2}$ yr$^{-1}$ in 2018 (Figure 2).

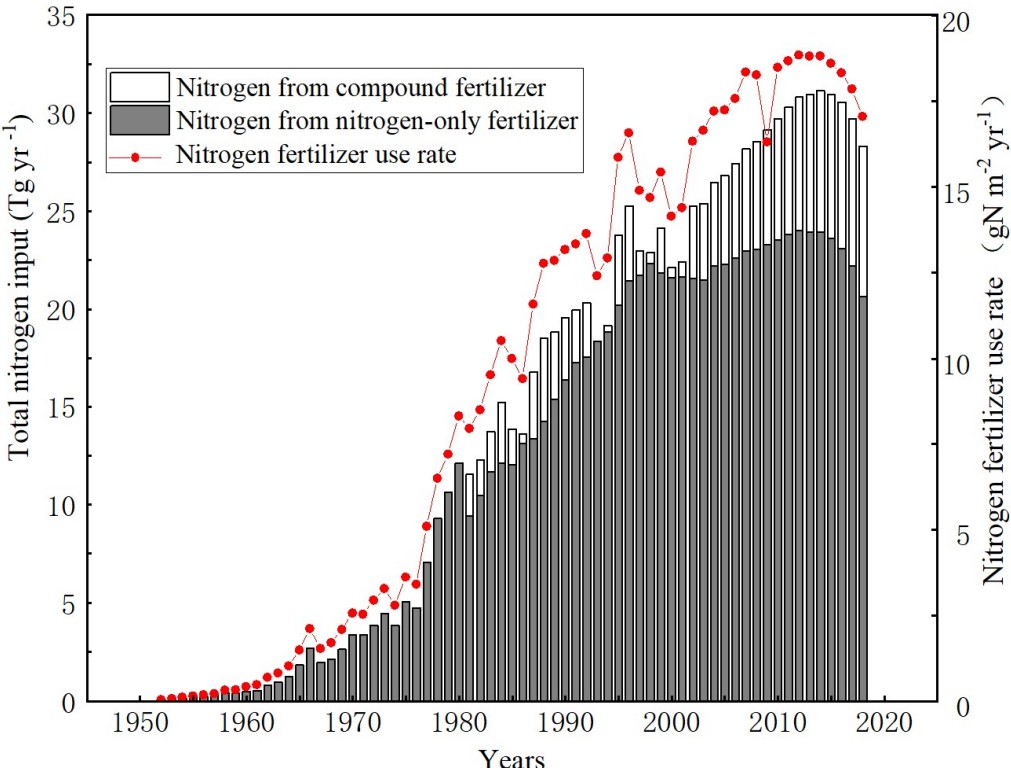

Figure 2. Total nitrogen fertilizers input and the average nitrogen fertilizers use rate in China's cropland from 1952 to 2018

## 3.2 Nitrogen fertilizers use by crop types in China

We examined the N fertilizers use in each of the major crop types planted in China since 1952 (Figure 3 is the average of different scenarios; the different scenarios were showed in Figure S2&3). The N fertilizers use rates were very low in 1950s and increased to 10-25 g N m$^{-2}$ yr$^{-1}$ in most crop types (Figure 3). Among all the crop types, soybean received the lowest N

inputs with the rate below 5 g N m$^{-2}$ (Figure 3). In comparison, extremely high N fertilizers use was observed in other crops (55-70 g N m$^{-2}$ yr$^{-1}$ in 2010s, Figure 3). Besides, vegetable received substantial amount of N fertilizers as high as 26 g N m$^{-2}$ yr$^{-1}$ in the 2010s then slightly decreased to 24 g N m$^{-2}$ yr$^{-1}$ in 2018 (Figure 3). The N fertilizers used rate in rice (i.e. early rice, mid-season rice, late rice) increased from the 1950s then leveled off in the mid-1990s, and the N fertilizers use rates for wheat, soybean, corn, oil seeds, cotton, and vegetable all kept increasing until the 2010s (Figure 3).

Corn is the only mono-crop receiving N fertilizers at approximately 7 Tg N yr$^{-1}$ in the 2010s (Figure 4 is the average of different scenarios; the different scenarios were showed in Figure S4&5). Among all the crop types, soybean received the least amount of N input which was about 0.1-0.2 Tg N yr$^{-1}$. The total N fertilizers used in soybean and oil seeds increased from the 1950s and then leveled off in the mid-1990s, and the N input for vegetable, mid-season rice, late rice, and wheat all kept

increasing until the 2010s (Figure 4). In comparison, the N fertilizers-use in cotton and other crops leveled off in the 1990s,
while the N input for early rice and late rice did not decline until the 1990s (Figure 4).

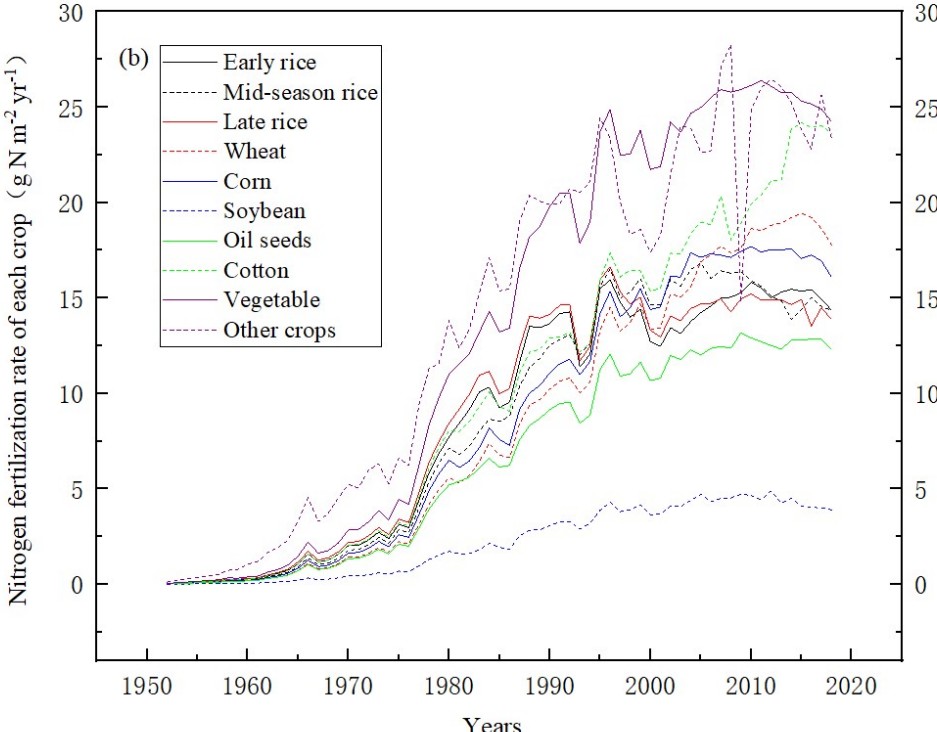

Figure 3. Nitrogen fertilizers use in major crop types from 1952 to 2018 (Unit: g N m$^{-2}$ yr$^{-1}$).

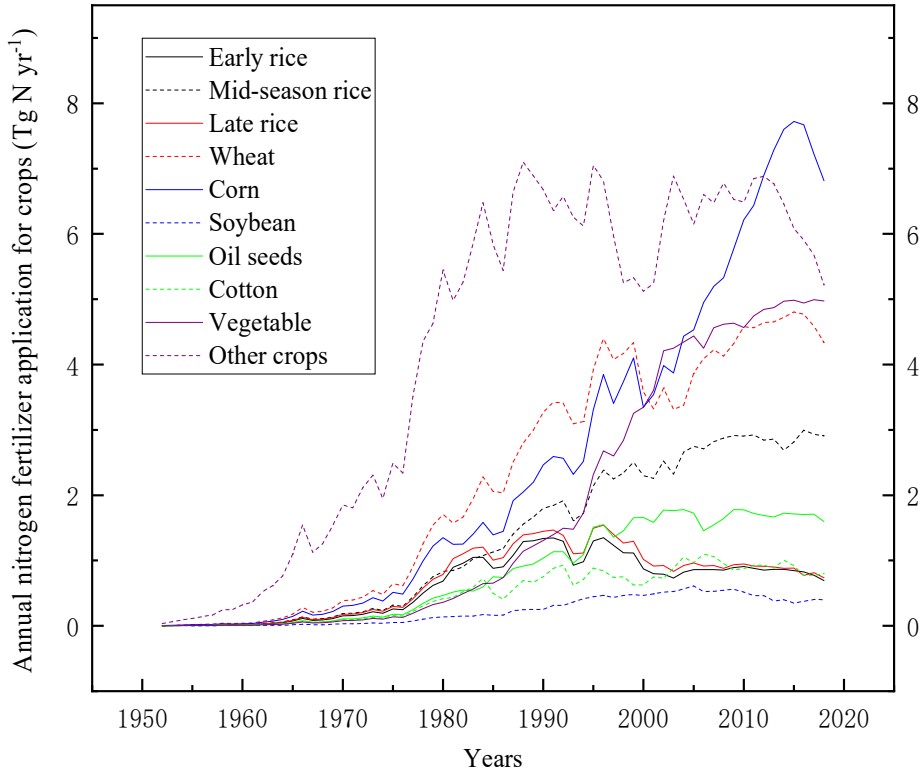

Figure 4. Total nitrogen fertilizer input in major crop types from 1952 to 2018 (Unit: Tg N yr⁻¹).

## 3.3 Spatial distribution of nitrogen fertilizers use in China

Our reconstructed maps show that the N fertilizers use in 1952 was generally lower than 1 g N m⁻² yr⁻¹ (Figure 5a), while the N fertilizers use increased to 4-8 g N m⁻² yr⁻¹ in 1980 and >16 g N m⁻² yr⁻¹ in 2000 in traditional agricultural plains including the Sichuan Plain and the Northern China Plain (Figure 5b). The higher N fertilizers use areas further expanded from 2000 to 2018 in the Northern China Plain, Northeast China Plain, and the northwest region (Figure 5d). In comparison, the uncertainties of the N fertilizers use rate were generally lower than 0.5 g N m⁻² yr⁻¹ in most of the area in China, but it kept increasing since 1952 to 2018 in northern, eastern, and northwestern regions (Figure 5g-h). Relatively higher uncertainties (> 4 g N m⁻² yr⁻¹) were found in the Northern China Plain in 2018 (Figure 5h).

We also showed the N fertilizer use rates in each of the major crop type in 2018 (Figure 6). Generally, the N fertilizer use was much higher in double-crop area than in monocrop area (Figure 6k vs Figure 6a-j). Moreover, early rice was seldom cultivated as monocrop in China (Figure 6a), which instead, was often planted with other crops in a year (part of the areas in Figure 6k). For monocrop areas, the N fertilizer use was found highest in corn (Figure 6e), while lowest N uses were detected in early rice and late rice (Figure 6a&6c).

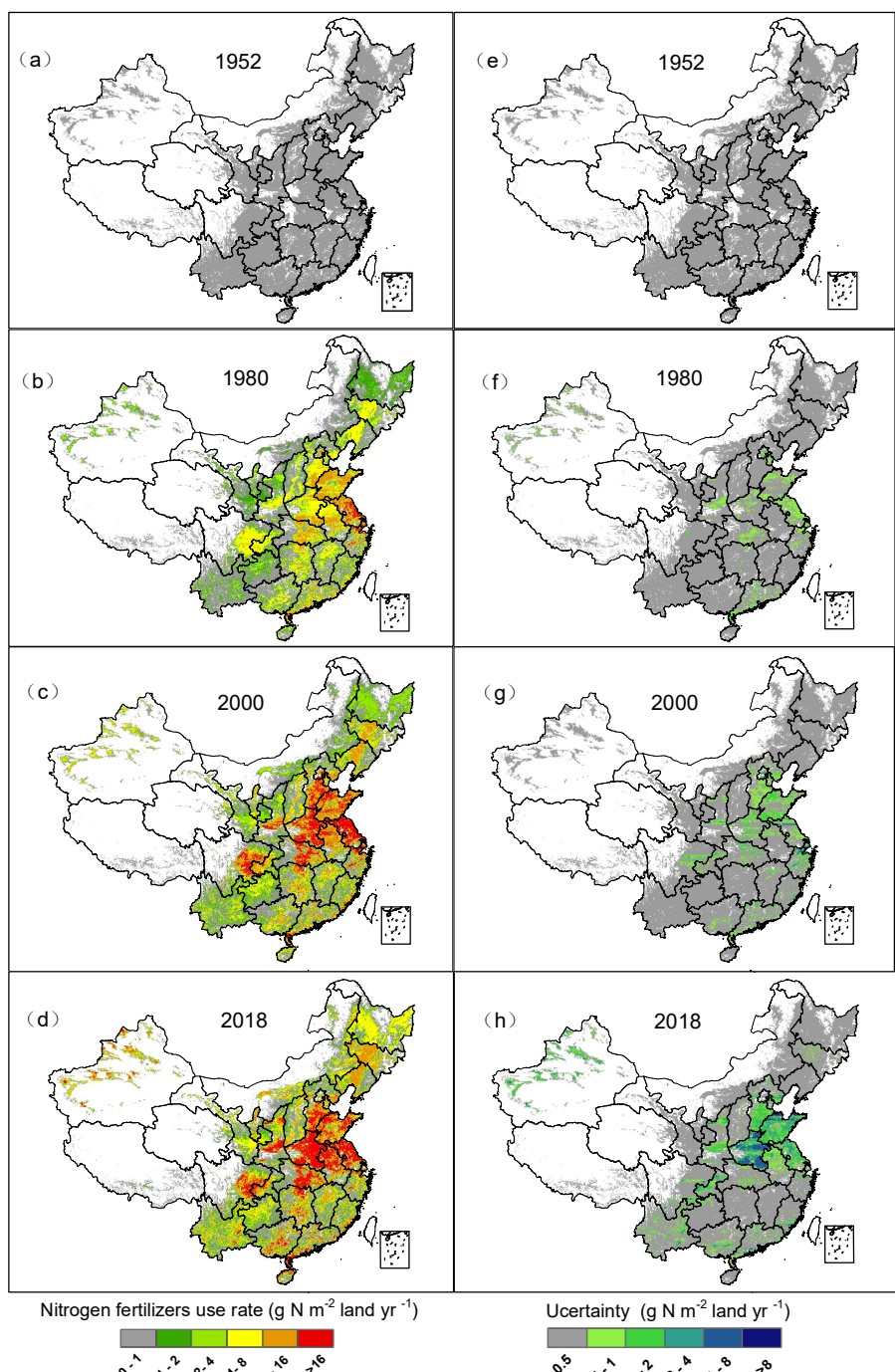

Figure 5. Spatial distribution of the rates (a-d) and the uncertainties (e-h) of nitrogen fertilizers use during different periods
215  in China (the four panels from the top to the bottom indicate the rates (left) and the uncertainties (right) in 1952, 1980, 2000,
and 2018, respectively; the value in the scale bar indicates the N fertilizers use rate per square meter of land).

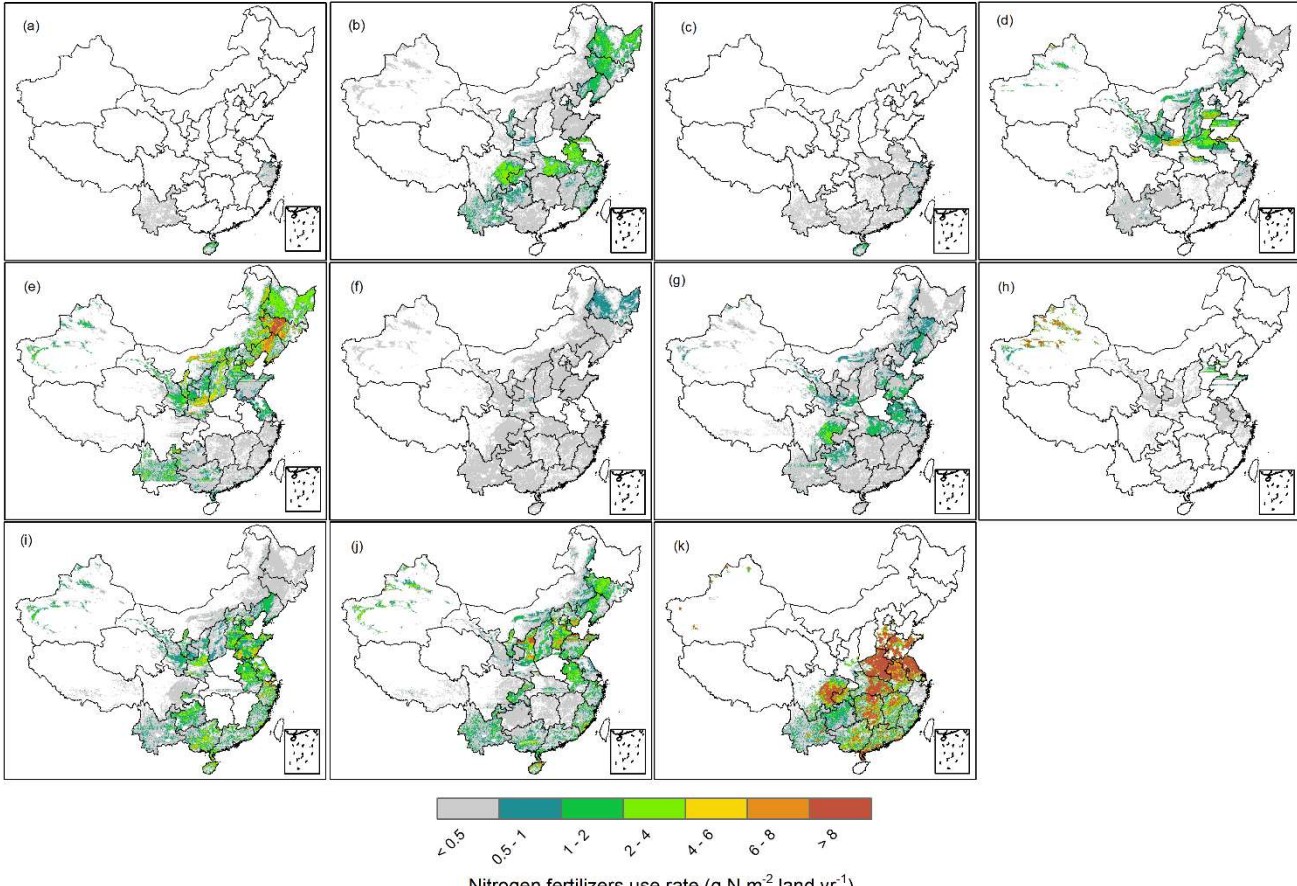

Figure 6. Spatial distribution of nitrogen fertilizer use rate by crop species in 2018 (panels a-h indicate the fertilizer use rate in early rice, mid-season rice, late rice, wheat, corn, soybean, oil seeds, cotton, vegetable, other crops, and double crops, respectively; the value in the scale bar indicates the N fertilizers use rate per square meter of land)

### 3.3 Comparison of newly reconstructed data and other data

We compared the reconstructed N fertilizers use rate maps with existing datasets. Specifically, the existing datasets describe N fertilizers use in two ways. The first one depicts N fertilizers use rate in per square meter cropland of each pixel (unit: g N m$^{-2}$ cropland yr$^{-1}$). We compared our reconstructed data with the datasets published by Nishina et al. (2017) and Lu and Tian (2017) in 1961, 1980, 1990, and 2005 or 2013 (Figure 7). The N fertilizer use in croplands (unit: g N m$^{-2}$ cropland yr$^{-1}$) of Nishina et al. (2017) was calculated by dividing the total N use maps (i.e. sum of NH$_4^+$ and NO$_3^-$ fertilizers) by the cropland coverage maps obtained from the LUHa (v1) (https://luh.umd.edu/luh_data). For the years 1961, all of the three datasets showed very low N fertilizer use (Figure 7a, 7e&7i). For 1980, our reconstructed data on N fertilizers use rates were generally

lower than the Nishina et al. (2017) and the Lu and Tian (2017)'s datasets in the most areas of China, except for the east, coastal provinces (Figure 7b, 7f&7j). For the year 1990, our reconstructed N fertilizer rates were generally higher than the other two datasets in the east of China (Figure 7c, 7g, 7k). For 2010, the N fertilizer use rates in raised to 20-35 g N m$^{-2}$ yr$^{-1}$ and >20 g N m$^{-2}$ yr$^{-1}$ in most of the study area derived from Lu and Tian (2017) and our study respectively (Figure 7h, 7l). Specifically, the N fertilizers use rates were lower in Lu and Tian (2017) than our reconstructed data in most of the northern

and eastern provinces in 2013 (provinces dominated by red color in Figure 8l).

The second approach we used N fertilizers use rates by per square meter of land (unit: g N m$^{-2}$ land yr$^{-1}$) for a comparison between our reconstructed data and other four studies by Potter et al. (2010), Nishina et al. (2017), Houlton et al. (2019), and Tian et al (2022) (Figure 8). Overall, we found similar patterns between our reconstructed data and other four datasets. Generally, other four datasets overestimated the China's cropland distribution in low-coverage areas, especially in

the north and northwest regions (grey area in Figure 8e-h). Besides, N fertilizers use rates in our reconstructed data were higher than the Potter et al. (2010) during the period of 1994-2001 in traditionally cultivated plains (e.g. the Sichuan Plain, the Northern China Plain, and the Northeast China Plain) (Figure 8a&e). Similar patterns were also found between the reconstructed map and Nishina et al. (2017)'s map in 2010, in which N fertilizer use was more concentrated in the traditionally cultivated plains (Figure 8b&f). When compared with the Houlton et al. (2019)'s dataset, our reconstructed data showed the

higher N fertilizers use in 2015 in the northwest region and the Northern China Plain, but lower N fertilizers use in the southwest region (Figure 8c&g). In comparison, the reconstructed N fertilizer use was generally higher than Tian et al (2022) across the entire study area (Figure 8d&h).

We also compared the total N fertilizer input at provincial derived from different studies in the period/years of 1994-2001, 2010, 2015, and 2018 (Figure 9). Generally, we found that the provincial N fertilizer use derived from our maps are

250 more close to the statistical results than data derived from other products as supported by higher $R^2$ (Figure 9). This indicates that our reconstructed maps are more advantaged than the existing data products in depicting the provincial nitrogen fertilizer use in China.

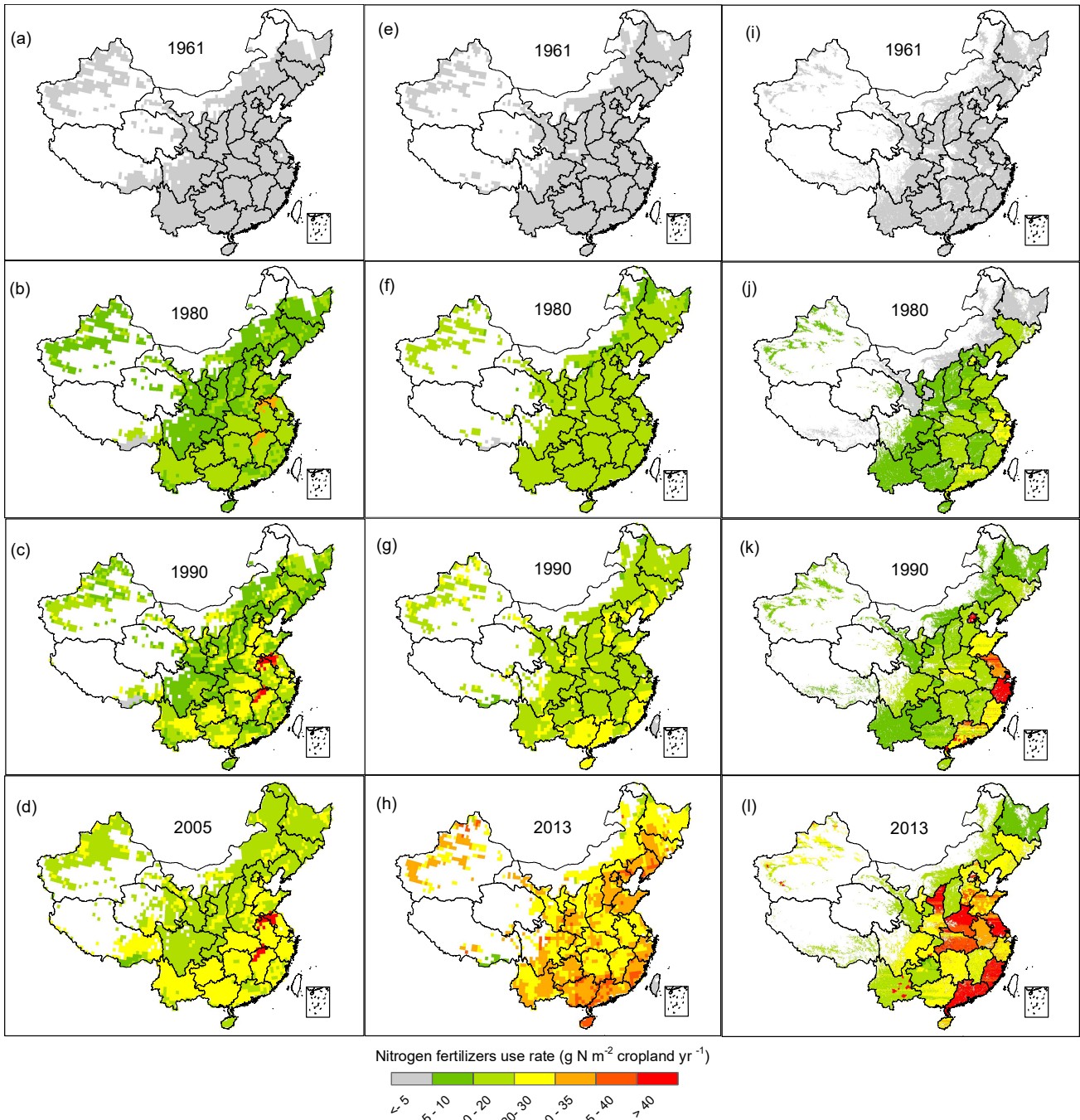

Nitrogen fertilizers use rate (g N m⁻² cropland yr⁻¹)

< - 5 | 5 - 10 | 10 - 20 | 20 - 30 | 30 - 35 | 35 - 40 | > 40

Figure 7. Distribution of nitrogen fertilizers use (a-c) in 1961, (d-f) in 1980, (g-i) in 1990, and (j-l) in 2005 or 2013 (left column: Nishina et al. (2017)'s data; central column: Lu et al. (2017)'s data; right column: this study; the values indicate N fertilizers use rates per square meter cropland of each grid-cell).

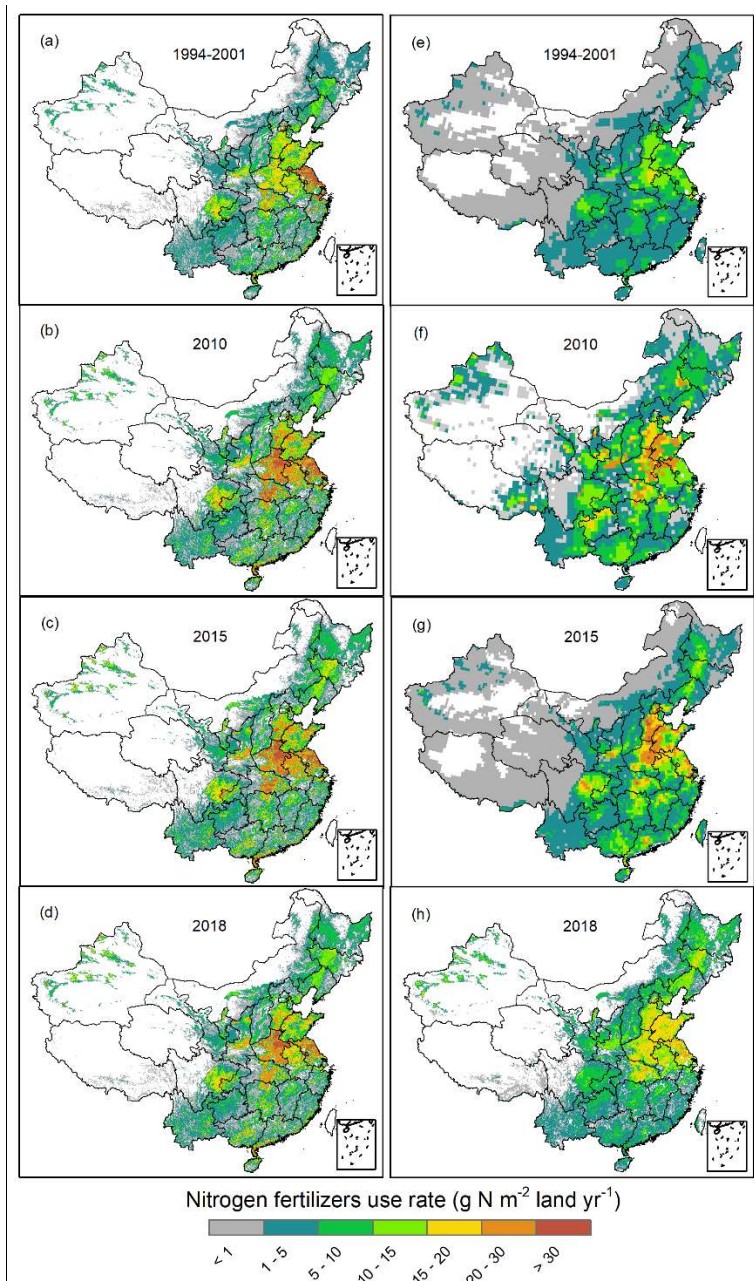

Nitrogen fertilizers use rate (g N m⁻² land yr⁻¹)

$< 1$  $1 - 5$  $5 - 10$  $10 - 15$  $15 - 20$  $20 - 30$  $> 30$

Figure 8. Comparisons of the nitrogen fertilizers use in different studies (the most recent year with both data from this study and other study available was used in comparing; panels a-d: this study: panels e-h: data from Potter et al. (2010), Nishina et al. (2017), Houlton et al. (2019) , and Tian et al (2022); the value indicates N fertilizers use rates per square meter of land).

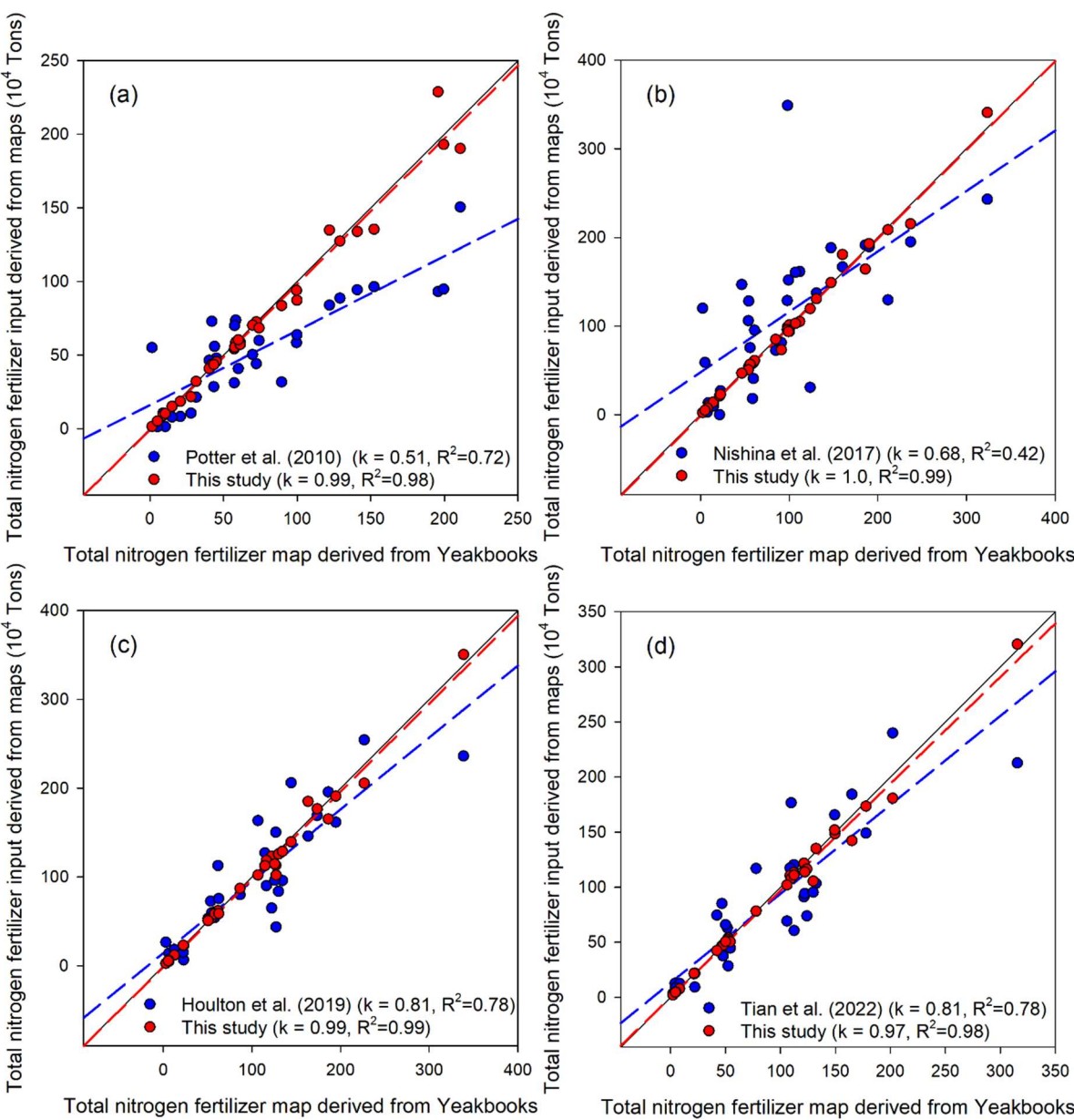

Figure 9. Comparisons of the total nitrogen fertilizer input in each province in China derived from different nitrogen fertilizer data products (the x-axis indicate the nitrogen fertilizer input obtained from the Chinese Statistical Yearbook; the black solid line indicate 1:1 line, and the red and blue lines indicate the linear regressions of the provincial nitrogen fertilizer input derive from this study and other studies; panels a-d show comparisons in the period/years of 1994-2001, 2010, 2015, and 2018, respectively; data of other studies were derived from the nitrogen fertilizer maps of Potter et al. (2010), Nishina et al. (2017), Houlton et al. (2019) , and Tian et al. (2022), respectively)

# 4 Discussion and conclusion

## 4.1 Temporal changes in the nitrogen fertilizers use

The temporal changes of the N fertilizers use in China's cropland during the study period (1952-2018) are unprecedented. The total N fertilizers input has increased by approximately 52 times from 0.6 Tg N yr$^{-1}$ in 1952 to the peak of 31.15 Tg N yr$^{-1}$ in 2014. Together with population growth, the cropland expansion and the N management in crop fields contributed to this sharp increase of N fertilizers use in China's cropland over time. Our study found 39.3% (41.1 Mha) increase of the area of planted crops from the 1950s to the 2010s (data derived from Chinese Statistical Yearbook), and 98 folds increase of the N fertilizers use, which were 0.19 g N m$^{-2}$ yr$^{-1}$ in the 1950s to 18.38 g N m$^{-2}$ yr$^{-1}$ in the 2010s suggesting a marked transformation of the China's environment.

The tremendous amount of N input in the soil has greatly altered the biogeochemical cycles and endangered terrestrial and aquatic ecosystems in China (Guo et al., 2010; Domagalski et al., 2007; Yang, 2001; Liu et al., 2014). Former studies claimed that there is a climate benefit potential of the N fertilizers use on soil carbon storage enhancement (Tian et al., 2011; Melillo et al., 2010). Yet, such benefits could be completely counteracted or even exceeded by the boosted N$_2$O production (Zaehle et al., 2011; Liu and Greaver, 2009). Lately, China has made greater efforts to reduce the N fertilizers use in croplands. For instance, the Ministry of Agriculture of China's "One control, two minus, three basic" policy for the "Fertilizer Use Zero-Growth Action Plan by 2020" greatly limited or even reduced the fertilizers use (The Ministry of Agriculture of China, 2015). Our study clearly identified the effect of this policy, in which we found the total N fertilizers input decreased by 9.1% (2.84 Tg N yr$^{-1}$) from 2014 to 2018 (Figure 2). Our results also revealed the reduction in the national N fertilizers use in cropland since 2014 was due to the less planted crop area and the better N management in crop fields. In our study, the crop planted area in China was decreased by 2.3% (3.4 Mha) from 2015 to 2018 (http://www.stats.gov.cn/) which consequently helped reducing the N fertilizer input in cropland by 0.63 Tg N yr$^{-1}$. The better N management (i.e. improved N fertilizers use) in crop fields was the significant factor for the reduction of national N input as evidenced by 9.4% reductions in the N fertilizers use rates from 2014 to 2018 in China's cropland that we identified in our study (Figure 2).

The dynamics of N fertilizers present in China's cropland is complex due to the nature of different fertilizers, their use and management practices. The decrease of the N fertilizers input since 2014 in our study was caused by the reduction of N-only fertilizers, while the N of the compound fertilizers has been increasing from 1993 to 2018 (Figure 2). It has been reported that because of the varying energy consumptions during production and application processes, some N present in compound fertilizers may increase (Chen et al., 2020; McLaughlin, 2000; Gellings and Parmenter, 2004). Besides, fertilizers response differently to environmental factors and are therefore varied in emission potentials (Bouwman et al., 2002; Shcherbak et al., 2014). Thus, the shift of the use of N fertilizer types might potentially affect the N$_2$O emission in both during the producing processes and during the field application, implying that this shift in fertilizers use habit can impact on GHG budget of China, which should be examined.

**4.2 Nitrogen fertilizers use by crop types**

Surprisingly, we found that the most intensively fertilized crop was not grain crop but vegetable and other crops in China (Figure 3). Some surveys documented that fertilizers received by vegetables could be 3.3 times higher than grain crops suggesting that vegetable and other crop lands would play different roles in GHG emissions than the grain crop (Huang et al., 2017; Hou et al., 2017). In our study, the major component of the other crops was orchard, which accounted for 10% of the total planted area in China (according to the data obtained from http://www.stats.gov.cn/) and indicated higher rates of N fertilizers use. We found about 2-to-5-folds of the N fertilizers use rates in orchards in China than in grain crops (Table 3). In our study, we found the highest N fertilizers use ($111.5-120.8$ g N m$^{-2}$ yr$^{-1}$) was in the apple orchard in the Loess Plateau (Table 3) suggesting that the orchards in dryland areas of China not only demand higher amount of water but also the increased use of N fertilizers. However, irrigation and higher N fertilizers use may together amplify the GHGs emissions as nitrification and denitrification processes might potentially be boosted (Trost et al., 2013). Given the large acreage of orchard and exponential rise of N$_2$O in response to N inputs increase (Bouwman et al., 2002; Shcherbak et al., 2014), it can be anticipated that the N$_2$O emission introduced by high N use in other crops might be an essential contributor to China's GHG budget. Many of the previous estimations have focused on GHG emissions from grain crops such as corn, rice, and wheat (Shcherbak et al., 2014; Ju et al., 2009; Huang and Tang, 2010), while our study implies that the GHGs emissions could be even larger from non-grain crops as the intensive N fertilizers use were identified in vegetable and other crops (Figure 3&4).

Despite we found the national average fertilizers use rates in grain crops (e.g. corn, rice and wheat) were reasonable, ranging from $12-20$ g N m$^{-2}$ yr$^{-1}$ in the 2010s in China (Figure 3), the rates greatly varied among provinces. This clearly indicates the spatial non-uniformity in N fertilizers use in grain crops and location-specific implications for soils and environments in China. For example, N-only fertilizers use in corn was as low as $2.2-7.3$ g N m$^{-2}$ yr$^{-1}$ in Northeast China Plain in 2018 known to be equipped with fertile soils (e.g. Jilin, Heilongjiang according to CSY data), while it was as high as $15.3-16.7$ g N m$^{-2}$ yr$^{-1}$ in less fertile soils in the northwest region in 2018 (e.g., Shannxi and Gansu according to CSY data). This is consistent to previous study, which revealed that 20%-40% of the grain crops were over-fertilized in China using national survey data collected from 2002-2005 (Yan et al., 2017). Although over-use of N fertilizers in grain crops has been widely reported in China (Wu et al., 2018; Zhang et al., 2015; Ren et al., 2021), our study indicates that the over-use might be more intensive in producing of vegetables and fruits, particularly in drier regions where demand of water and fertilizers is high. Former study also suggested that the vegetables and fruits had 2.4-6.2 times higher N fertilizers demand in China ($38.8-55.5$ g N m$^{-2}$ yr$^{-1}$) than in Europe and the U.S. ($9.6-16.5$ g N m$^{-2}$ yr$^{-1}$), resulting into a much lower N fertilizers use efficiency in these crops (1/2-1/3 from that of the U.S.) (Wu et al., 2016). Since vegetables and orchard area accounts for 20% of the total planted area in China (CAY data in 2018), such high N fertilizers use rates together with the water use imply a large potential of economic and environmental benefits from optimizing fertilizers use in these non-grain crops.

Table 3 Nitrogen fertilizers use rates in minor crops reported in China.

| Crop type | N fertilizers use (g N m$^{-2}$ yr$^{-1}$) | References |
|---|---|---|
| Barley | 22.9 | (Zaituniguli et al., 2021) |
| Sugarcane | 36.0 | (Zong, 2017) |
| Tobacco | 51.8 | (Hou et al., 2017) |
| Sugar crop | 69.6 | (Hou et al., 2017) |
| Apple (Loess Plateau) | 111.5-120.8 | (Chen et al., 2018) |
| Apple (Shandong) | 49.0 | (Wei and Jiang, 2012) |
| Peach | 55.0 | (Gao, 2010) |
| Citrus | 51.3 | (Liang, 2007) |
| Grape (Northern China) | 114.2 | (Lu et al., 2012) |
| Pear | 55.5 - 68.7 | (Wu et al., 2016; Zhang et al., 2020) |

## 4.3 Spatial trends of nitrogen fertilizers use

China, being a large country with varying geographic and climatic features, the N fertilizers use rates in crops are dependent largely on types of the crop and the location where the crops are grown. In addition, how the cropland is being managed also plays a greater role in N fertilizers use rates. Previous studies revealed that the optimal level of the N fertilizers use rates for crops were at around 15-20 g N m$^{-2}$ yr$^{-1}$ in China (Zhu and Chen, 2002; Ju et al., 2009, 2004). In our study, the N fertilizers use rates maps revealed some of the widespread, over-fertilized crop field regions in China (Figure 7l). Surprisingly, the most over-fertilized provinces, which majorly distributed in the central China and the southeast China, received more than two-folds N fertilizers use rates than the suggested optimal rates (Figure 7l).

The N fertilizers use rates have become an important driver directly determining the estimations of the crop productions and the GHG emissions accounting in different parts of the world (Tian et al., 2018). Hence, the reconstructed N fertilizers use data in our study is potentially useful to improve the accounting of the GHGs budget in China, especially the N$_2$O emissions. In a previous study, we found that the widely used, FAO-based global land use and land cover data (e.g. HYDE, LUH2) overestimated the cropland distribution in low coverage areas but underestimated the cropland percentage in high coverage areas (Yu et al., 2021). This might explain the differences in the N fertilizers use rates between our reconstructed data and other products such as Potter et al. (2010), which were being released previously. When we presented the FAO-based cropland maps (Figure 8e, h), on the N fertilizers use product of the Potter et al. (2010) and Tian et al. (2021), with

overestimated low-coverage croplands in China, the maps clearly reflected diluted, low N fertilizers use rates in more intensively cultivated plains. Recently, the International Modeling Community has initiated a global $N_2O$ Model Intercomparison Project (NMIP) to quantify the $N_2O$ gas emission from the land surface using ten state-of-the-art terrestrial biosphere models (Tian et al., 2018), where the HYDE cropland and HYDE-based N fertilizers use data were selected as the forcing data in simulation of underrated/overrated $N_2O$ emissions in high/low cropland coverage areas (Lu and Tian, 2017; Tian et al., 2018; Wang et al., 2020a). We consider that our reconstructed N fertilizers use data will be vital for quantifying the $N_2O$ gas emissions from China's croplands in future and substantially help mitigate global warming.

We would like to reiterate that the provincial N fertilizers use rates reported in the Cost-benefit Report of the National Agricultural Products (CBR) were the most important reference we adopted while reconstructing the N fertilizers use maps in this study. Nonetheless, for each province, there were cases, where always a few crop types were thinly planted, but missed the reporting of the N fertilizers use rates. For those missing crop types reported in each province, we filled the gap by assuming the N fertilizers use rates close to the nearby province or equal to the national average (see the method section). However, the N fertilizers use rates might greatly differ from our assumptions because farmers' habits largely varied. Besides, the N fertilizers input contributed from compound fertilizers were not directly available, which also introduced uncertainty in the total N fertilizers applied for each crop. Thus, we could have improved the N fertilizers use maps if we had obtained more detailed surveying data. The distribution of the crop types which missed reporting at the sub-county levels and some years increased uncertainty of the reconstruction further. We derived the crop rotation information from national surveys of 2341 counties carried out in 1980, 1990, 2000, 2002, and 2011 (Liu et al., 2018), as these data were not sufficient, so that we assumed that the crop types planted between surveyed years were relatively stable and the information of the nearest year was used as substitution. Therefore, there might have been higher uncertainties in the N fertilizers use rates reconstruction in non-surveyed years. Despite such limitations, our reconstructed data, developed from improved gridded maps and crop-specific statistical information, are advantageous in depicting the N fertilizers use and estimating the GHG ($N_2O$ emission) budgets in China's croplands.

## 5 Data Availability

The N fertilizers use maps and the crop-specific N fertilizer use maps reconstructed in this study are publicly available via https://doi.org/10.6084/m9.figshare.21371469.v1 (Yu, 2022).

## Author contributions

ZY designed the work. JL collected data and performed data entry. ZY analysed the data and wrote the paper. GRK provided comments to improve the manuscript.

**Competing interests**

The authors declare that they have no conflict of interest.

**Acknowledgement**

We greatly acknowledge data entry assistance from Yanli Dong. We thank Dr. Kazuya Nishina and the two anonymous reviewers for their constructive suggestions for improving the manuscript. This work was supported by National Natural Science Foundation of China (No. 32001166), the Natural Science Foundation of Jiangsu Higher Education Institution of China (20KJB170013), and the Startup Foundation for Introducing Talent of NUIST (No. 2019r059).

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
