# Peer review of "Historical nitrogen fertilizers use in China from 1952 to 2018"

_Earth System Science Data, 2022_

## Referee Comment (RC1)

**Review: essd-2022-263**

Dear authors

   This study, entitled "Historical nitrogen fertilizers use in China from 1952 to 2018", tried to develop a new detailed map of nitrogen fertilizer inputs in China. It is within the scope of the ESSD. It contains some new perspectives (crop-specific, crop rotation, etc.) and is basically worthy of publication, but it requires for some corrections before the publication. Some of the descriptions regarding the process of making the data set are unclear. Major comments are below on the points that need to be corrected.

**General comments**

1. First of all, thank you very much for citing Nishina et al. (2017) (Nishina map from here). However, please note that the Nishina map is slightly misinterpreted in your comparison. As it uses LUHa (Hurtt et al., 2011) as a map of agricultural land, so it does not consider N inputs where LUHa has 0% agricultural land area. This means that there are at least some areas with zero at all in Nishina map. Therefore, Figure 6 is incorrect. In addition, Nishina map consider the double cropping region, which is based on the crop use intensity (CUI) map provided by Siebert et al. (2010), In the double cropping region, it is needed to be twice the annual input. Otherwise, the FAO input cannot be reproduced. Please refer to Figure 1 in Nishina et al. (2017), which illustrates where we treated as two-season crop areas (map for double cropping areas in Nishina map can also be shared). As for the annual nitrogen input per half-degree grid cell, if you add up the individual fertilizer input data for NH4+ and NO3- by month, and then aggregate that data for each year. After this, the N input per unit area of cropland can be calculated by dividing by the area of farmland in LUHa. I think you can create a comparable map.

   Nevertheless, this issue has nothing to do with the quality of your dataset.

2. As a matter of fact, even if you use Global maps for comparison, there are better maps available for China. One is under review, but I can refer you to two papers by Wang et al. (2020) or Tian et al (under review in ESSD). Please consider comparing with this one.

3. The method of Gap-filling is not clear. Especially when Gap-filling is applied to space, the total amount of nitrogen fertilization may be larger than the statistics used, depending on the method.

4. On a related point, each and every procedure should be formulated in mathematical formulas (even with respect to simple tabulations). Everything is described by text, making it difficult to see the validity of the method. The same applies to the description of Gap-filling. As other example, I could not see how to calculate and define uncertainty (shown in Fig 5) in the current manuscript.

5. Please add units to all drawings. Indeed, different units are mixed (e.g., "gN per unit land per year", "N fertilizers use rate per square meter of cropland ") in the text. So, it is not clear at first glance what you are referring to in the figure.

**Reference**

**Wang et al (2020)** Data-driven estimates of global nitrous oxide emissions from croplands. Natl. Sci. Rev. 7 441201352

**Tian et al (under review)** A 5-arcmin resolution annual dataset from 1860 to 2019. ESDD [preprint], https://doi.org/10.5194/essd-2022-94

---

## Author Comment (AC3)

**Response to reviewer comments**

We thank the reviewer for the precious and constructive suggestions to improve our manuscript. Please find our point-by-point response below.

**Reviewer 3:**

This work reconstructed a historical, annual N fertilizers use dataset in China at 5 km  $\times$  5 km resolution covering the period of 1952 to 2018 by integrating improved cropland maps. The dataset is useful for many purposes. Generally the paper is well-written, I have several concerns for authors to improve it before accepting for publication.

Response: We thank the reviewer for valuing our work! We have addressed all the concerns raised by the reviewer.

1) The Figure 1. Methodology flowchart of nitrogen (N) fertilizer map reconstruction is not detailed enough to illustrate the methods used. Need to be improved by inclusing the major methods used.

Response: We thank the reviewer for the suggestion. We agree and revised the methodology flowchart to help clear the methods used. Specifically, we added the temporal coverage of the data, the details of the processes, and the methods implemented. Please check our new flowchart below:

Figure 1. Methodology flowchart of nitrogen (N) fertilizer map reconstruction

2) Are the spatial pattern of N fertilizers use in each of the major crop types planted in China available? Why not resported in the main text? Are the data are availbale? Response: We thank the reviewer for this precious suggestion. We have calculated the N fertilizer use in each of the major crop types in China. These data will be released together with the original N input product during the next step of manuscript reviewing process. Also, discussion about this information will be added in the main text. Here is an example showing the spatial distributions of N fertilizer use rate by crop types in China in 2018 (note the panel h indicate the N fertilizer use rate of double crops):

---

## Author Response (AR1)

**Response to reviewer comments**

We thank the three reviewers and the editor for the precious and constructive suggestions to improve our manuscript. We have addressed all the comments raised. Please find our point-by-point response below.

**Editor:**

Please upload all related data, particularly those crop-specific N fertilizer use maps.

Response: We thank the reviewer for the suggestion. We have uploaded all data, and specifically, the crop-specific N fertilizer maps have been deposited in Figshare together with the N fertilizer data (https://doi.org/10.6084/m9.figshare.21371469.v1).

**Reviewer 1:**

This study, entitled "Historical nitrogen fertilizers use in China from 1952 to 2018", tried to develop a new detailed map of nitrogen fertilizer inputs in China. It is within the scope of the ESSD. It contains some new perspectives (crop-specific, crop rotation, etc.) and is basically worthy of publication, but it requires for some corrections before the publication. Some of the descriptions regarding the process of making the data set are unclear. Major comments are below on the points that need to be corrected.

Response: We thank the reviewer for valuing our work! The suggestions are very helpful for improving the manuscript. Below we have made the corrections as per his suggestions.

**General comments**

1. First of all, thank you very much for citing Nishina et al. (2017) (Nishina map from here). However, please note that the Nishina map is slightly misinterpreted in your comparison. As it uses LUHa (Hurtt et al., 2011) as a map of agricultural land, so it does not consider N inputs where LUHa has 0% agricultural land area. This means that there are at least some areas with zero at all in Nishina map. Therefore, Figure 6 is incorrect. In addition, Nishina

map consider the double cropping region, which is based on the crop use intensity (CUI) map provided by Siebert et al. (2010), In the double cropping region, it is needed to be twice the annual input. Otherwise, the FAO input cannot be reproduced. Please refer to Figure 1 in Nishina et al. (2017), which illustrates where we treated as two-season crop areas (map for double cropping areas in Nishina map can also be shared). As for the annual nitrogen input per halfdegree grid cell, if you add up the individual fertilizer input data for NH4+ and NO3- by month, and then aggregate that data for each year. After this, the N input per unit area of cropland can be calculated by dividing by the area of farmland in LUHa. I think you can create a comparable map.

Nevertheless, this issue has nothing to do with the quality of your dataset.

Response: We thank the reviewer for pointing out the issues in Figure 6. We have updated the Figure 6 with total N use annually aggregated from the monthly $NH_4^+$ and $NO_3^-$ input data. Besides, we also derived the N input per unit area of the cropland using the approach suggested. Specifically, we downloaded LUHa data from https://luh.umd.edu/luh_data/. The data (LUHa.v1) covers the period of 1500 to 2005, and the years 1961, 1980, 1990, and 2005 (the most recent year is 2005 in LUHa.v1 data) were used for calculating the nitrogen fertilizer use rate per unit cropland area (see Figure 7). If the updated version of LUH data is more appropriate (e.g. LUH2), we are also happy to further update the calculations.

Besides, we also used the Nishina et al. (2017) data for the comparison of nitrogen fertilizer use rate at per square meter of cropland in 2010. Please also see the updated comparisons in Figure 8.

Here are the updated figures:

The updated Figure 7 (i.e. Figure 6 in the original version):

[Figure]

Figure 7. Distribution of nitrogen fertilizers use (a-c) in 1961, (d-f) in 1980, (g-i) in 1990, and (j-l) in 2005 or 2013 (left column: Nishina et al. (2017)'s data; central column: Lu et al. (2017)'s data; right column: this study; the values indicate N fertilizers use rates per square meter cropland of each grid-cell).

The updated Figure 8 (i.e. Figure 7 in the original version):

[Figure]

Figure 8. Comparisons of the nitrogen fertilizers use in different studies (the most recent year with both data from this study and other study available was used in comparing; panels a-d: this study: panels e-h: data from Potter et al. (2010), Nishina et al. (2017), Houlton et al. (2019), and Tian et al (2022); the value indicates N fertilizers use rates per square meter of land).

2. As a matter of fact, even if you use Global maps for comparison, there are better maps available for China. One is under review, but I can refer you to two papers by Wang et al. (2020) or Tian et al (under review in ESSD). Please consider comparing with this one.

Response: We thank the reviewer for the suggestion. We agree and have added the map comparisons with Tian et al (2022)'s data as suggested (please see changes made in Figure 7). We also contacted Wang et al. (2020) asking for the data, but it was not available since they will publish the data first.

3. The method of Gap-filling is not clear. Especially when Gap-filling is applied to space, the total amount of nitrogen fertilization may be larger than the statistics used, depending on the method.

Response: We thank the reviewer for pointing out this. We realize that our descriptions about gap-filling (i.e., temporal gap-filling and spatial gap-filling) were misleading. In this study, temporal gap-filling and spatial gap-filling were both performed at rebuilding the provincial, crop-specific N fertilizer use table. Specifically, the table describes the N fertilizer use rate for each of the 10 major crop types in each province from 1952 to 2018. We first allocated the N fertilizer use rates obtained from the Cost-benefit Report of the National Agricultural Products (CBR) reports. There were many missing data for certain years and crop types. If a crop type in a province had N fertilizer use rate reported in the CBR, gap-filling of the missing years were treated as "temporal gap-filling". If a crop type in a province had N fertilizer use rate never reported, then we treated the gap-filling as "spatial gap-filling" (because the N use rate was derived from national average or nearby provinces). Specifically, the "spatial gap-filling" was a process to allocate N use rates for crops planted in a province (areas are usually very small) but their N fertilizer use were never reported in the province (usually minor crops for the province). Therefore, both gap-fillings were done in the table, and the gap-filled N use rates were further adjusted to maintain the total N input close to national statistics.

We realized these descriptions might be confusing. To avoid misunderstanding, we have rephrased this part by abandoning the use of the terms "temporal gap-filling" and "spatial gap-filling". Here is the revised text (Please check our revisions in Lines97-106):

"The CBR data provides officially released fertilizers use information summarized from thousands of samples collected in each province in China. First, we created an empty table to record the N fertilizer use rate for each province with all the 10 types included. Second, the N use rate of the table was allocated using data obtained from CBR when available. Third, if a crop type was never planted in the province, the N use rate was set to 0. Fourth, we checked and gap-filled the missing N fertilizers use rates in the province. For crop type with N use rate intermittently reported, we linearly interpolated the rate using the two nearest data reported before and after the year (see equation 2). While for crop type been planted in the province but its N fertilizer use were never reported, two fertilizers use scenarios were considered."

4. On a related point, each and every procedure should be formulated in mathematical formulas (even with respect to simple tabulations). Everything is described by text, making it difficult to see the validity of the method. The same applies to the description of Gap-filling. As other example, I could not see how to calculate and define uncertainty (shown in Fig 5) in the current manuscript.

Response: We thank the reviewer for the suggestion. We agree and have added formulations for describing the method. Please check our revisions in Lines76-146. Specifically, for the uncertainty analysis, the it was derived from the eight scenarios considered. To make it clearer, we have added a table to clarify further.

5. Please add units to all drawings. Indeed, different units are mixed (e.g., "g N per unit land per year", "N fertilizers use rate per square meter of cropland ") in the text. So, it is not clear at first glance what you are referring to in the figure.

Response: We thank the reviewer for the suggestion. We have added units to all figures as suggested. Please check our revisions in Figures 7&8 before and Figure 5 below:

[Figure]

Figure 5. Spatial distribution of the rates (a-d) and the uncertainties (e-h) of nitrogen fertilizers use during different periods in China (the four panels from the top to the bottom indicate the rates (left) and the uncertainties (right) in 1952, 1980, 2000, and 2018, respectively; the value in the scale bar indicates the N fertilizers use rate per square meter of land).

**Reviewer 2:**

It is very valuable to estimate the crop-specific fertilizer in China. However, there are some serious problems in the process of accounting for crop-specific fertilizer, so I question the quality of the final data product.

Response: We thank the reviewer for the suggestions. We fully respect the opinion of the reviewer. We have made a rebuttal by addressing each concern raised in the manuscript below.

1. For the fragmented crop distribution in most parts of southern China, this paper calculated the crop-specific fertilizer application rate from 1952 to 2018 based on the cropland area data with a resolution of 5 km produced by provincial data and just five years' crop rotation data. I have serious doubts about the accuracy of the data.

Response: We thank the reviewer for the suggestion. There are some misunderstandings here and we would like to make these clear. First, the crop-specific fertilizer application rate was originally allocated to a 100-m crop rotation map and resampled to 5-km (not directly allocated to 5 km cropland data). The 100-m crop distribution maps were developed in previous study (an intermediate product before resampled to 5-km cropland maps, please see Yu et al. 2021). Therefore, the fertilizer map was not directly developed at 5-km resolution. Second, the rotation map used for N fertilizer rate allocation were different in each year (not the five fixed maps). The five county-level rotation maps served as potential rotation maps when allocating crop type spatially. For example, in 1981, the nearest-year, county-level rotation maps (e.g. 1980 map) were used as potential rotation map to allocate each crop types spatially. Specifically, a cropland grid-cell was given priority to be allocated the crop type found in the corresponding grid-cell from the potential rotation map in the nearest year. However, the cropland map varies between years, resulting into the dynamics of the planted area annually. Therefore, the rotation map of a different year will also need to be adjusted to ensure the planted area of each crop type to be equal to the data

from the officially released reports. The uncertainty has also been clearly discussed in the main text (please see the last paragraph of the discussion section).

Similar approach was adopted in our former study (Cao, Lu, Yu 2018), in which only the rotation data since 2008 was available (please see CDL maps here: https://www.nass.usda.gov/Research_and_Science/Cropland/Release/). However, based on the state-level inventory data (similar to provincial data in China), we extended the rotation maps from 2008 back to 1850 (please see section 2.1.1 in Yu et al. 2018). Despite a limited and shorter coverage of the observed data (e.g. CDL rotation maps from 2008 to 2015), our reconstructed N fertilizer data at 5 arc min × 5 arc min (~8 km resolution) greatly improved the biogeochemical simulations, including $N_2O$ emission accounting (Lu et al. 2022), crop production evaluation (Lu et al. 2018), and carbon budget assessment (Yu et al. 2019). These are strong supports that using a longer coverage data at provincial level in China is decent and reliable (e.g. rotation maps cover the period of 1980 to 2011 at county level).

About the accuracy of the data, please also see our response to the second questions below.

References:

Cao, P., Lu, C., & Yu, Z. (2018). Historical nitrogen fertilizer use in agricultural ecosystems of the contiguous United States during 1850–2015: application rate, timing, and fertilizer types. *Earth System Science Data*, *10*(2), 969-984.

Lu, C., Yu, Z., Zhang, J., Cao, P., Tian, H., & Nevison, C. (2022). Century-long changes and drivers of soil nitrous oxide (N2O) emissions across the contiguous United States. *Global Change Biology*, *28*(7), 2505.

Lu, C., Yu, Z., Tian, H., Hennessy, D. A., Feng, H., Al-Kaisi, M., ... & Arritt, R. (2018). Increasing carbon footprint of grain crop production in the US Western Corn Belt. *Environmental Research Letters*, *13*(12), 124007.

Yu, Z., Jin, X., Miao, L., & Yang, X. (2021). A historical reconstruction of cropland in China from 1900 to 2016. *Earth System Science Data*, *13*(7), 3203-3218.

Yu, Z., Lu, C., Cao, P., & Tian, H. (2018). Long-term terrestrial carbon dynamics in the Midwestern United States during 1850–2015: Roles of land use and cover change and agricultural management. *Global Change Biology*, *24*(6), 2673-2690.

Yu, Z., Lu, C., Tian, H., & Canadell, J. G. (2019). Largely underestimated carbon emission from land use and land cover change in the conterminous United States. *Global Change Biology*, *25*(11), 3741-3752.

2.    When calculating crop-specific fertilizer application, the article mentioned "The N fertilizers use rate for each major crop types (except other crops) was intermittently reported in the Cost-benefit Report of the National Agricultural Products (CBR) covering the period of 2004-2018 (Table 1)". However, there is no corresponding crop fertilizer allocation table in the text or supporting materials, nor is there a link to the data source. I entered to the CBR website to check, but did not get the corresponding data. And this part of data is very critical, which directly affects the accuracy of the final product. Moreover, so-called high-resolution data, based only on provincial rates of crop fertiliser allocation, are crude.

Response: We are sorry that the source for CBR data information was not provided in our original submission. The data can be obtained from the following link: https://data.cnki.net/trade/Yearbook/Single/N2021120200?zcode=Z009

     We have added this information in the revised main text. The CBR data was published in Chinese, and a sample data table is pasted below by showing the fertilizer use for corn in 2007 in a few number of provinces:

**Fertilizer use for corn**

**2-6-3 2006 年各地区玉米化肥投入情况**

| 项 目 | 单位 | 平 均 | 北 京 | 天 津 | 河 北 | 山 西 | 内蒙古 |
|---|---|---|---|---|---|---|---|
| 一、每亩化肥金额 | 元 | 85.20 | 75.96 | 84.69 | 71.00 | 75.48 | 93.64 |
| （一）氮肥 | 元 | 42.13 | 45.96 | 43.99 | 34.78 | 36.05 | 47.22 |
| 1.尿素 | 元 | 34.21 | 44.70 | 42.78 | 30.91 | 19.47 | 42.47 |
| 2.碳铵 | 元 | 7.40 | 1.26 | 1.21 | 3.80 | 15.94 | 4.75 |
| 3.其他氮肥 | 元 | 0.52 | | | 0.07 | 0.64 | |
| （二）磷肥 | 元 | 3.27 | | | 0.69 | 14.11 | 0.58 |
| 其中:过磷酸钙 | 元 | 2.78 | | | 0.27 | 12.88 | |
| （三）钾肥 | 元 | 1.20 | | | 0.02 | 2.86 | |
| 其中:氯化钾 | 元 | 0.73 | | | 0.02 | 0.08 | |
| （四）复混肥 | 元 | 38.19 | 30.00 | 40.70 | 35.44 | 22.25 | 45.56 |
| 1.复合肥 | 元 | 35.73 | 30.00 | 40.70 | 33.11 | 17.97 | 45.56 |
| 其中:二铵 | 元 | 10.49 | 13.64 | 40.70 | 14.00 | 1.05 | 29.46 |
| 2.混配肥 | 元 | 2.46 | | | 2.33 | 4.28 | |
| （五）其他肥料 | 元 | 0.41 | | | 0.07 | 0.21 | 0.28 |
| 二、每亩化肥折纯用量 | 公斤 | 20.05 | 17.89 | 20.56 | 17.08 | 20.29 | 21.61 |
| （一）氮肥 | 公斤 | 10.80 | 11.32 | 11.28 | 9.05 | 10.01 | 12.03 |
| 1.尿素 | 公斤 | 8.34 | 10.91 | 10.91 | 7.80 | 4.74 | 10.46 |
| 2.碳铵 | 公斤 | 2.31 | 0.41 | 0.37 | 1.23 | 5.14 | 1.57 |
| 3.其他氮肥 | 公斤 | 0.15 | | | 0.02 | 0.13 | |
| （二）磷肥 | 公斤 | 1.10 | | | 0.26 | 4.27 | 0.05 |
| 其中:过磷酸钙 | 公斤 | 0.93 | | | 0.09 | 3.84 | |
| （三）钾肥 | 公斤 | 0.28 | | | | 0.70 | |
| 其中:氯化钾 | 公斤 | 0.18 | | | | 0.03 | |
| （四）复混肥 | 公斤 | 7.87 | 6.57 | 9.28 | 7.77 | 5.31 | 9.53 |
| 1.复合肥 | 公斤 | 7.39 | 6.57 | 9.28 | 7.27 | 4.28 | 9.53 |
| 其中:二铵 | 公斤 | 2.35 | 3.23 | 9.28 | 3.25 | 0.24 | 6.53 |
| 2.混配肥 | 公斤 | 0.48 | | | 0.50 | 1.03 | |

Indeed, the CBR data is one of the most critical data available for research which may determine the accuracy of the final product. However, the CBR is the legitimate data source (please see introduction of the 2019 report in this link: http://www.stats.gov.cn/tjsj/tjcbw/202008/t20200824_1785455.html), which provides officially released fertilizers use information summarized from thousands of samples collected in each province in China. The first CBR was published in 1981, while the crop-specific fertilizer use was not available until 2004. We purchased the reports and entered the data manually for our study. As the CBR data source has become the most reliable and available data in China, we do not agree that provincial data are crude.

First, the provincial, crop-specific N fertilizer use rate obtained from CBR are reliable (supported by large sample collected during summarization as aforementioned). This guarantees the accuracy of provincial results. Second, at a finer scale (i.e., sub-provincial), the N fertilizer use is more closely related to crop type planted spatially (e.g., corn vs soybean). Therefore, sub-provincial N fertilizer use pattern were determined by crop rotation maps. The crop rotation maps, developed from county-level survey data (2341 counties, please check Liu et al. (2018) for more details), are also the most reliable maps available in China to date. Third, the rotation maps were also dynamic as adjusted by the planted area of each crop type officially reported (please also see our response before). Moreover, comparing with former N fertilizer product developing from crop specific fertilizer use from country level (e.g., Lu and Tian 2017), our provincial data is a step forward in providing a finer N fertilizer use data in China.

Despite there are uncertainties (and we admit it), we have explained this in the last paragraph of the discussion of the manuscript (Lines373-384 in the revised manuscript). Another advantage of our data is that we developed the N fertilizer use maps based on improved cropland data. As elaborated in our previous studies (Yu et al. 2021, Yu et al. 2022), FAO-based data greatly biased in depicting cropland distribution in China (see Figures 1&2 below). The two most serious biases are: 1) FAO-based cropland data underestimated cropland coverage in traditional cultivated areas, but it overestimated cropland coverage in low cultivated areas (see Figure 1a-e and 1i-m below); and 2) the temporal change of cropland coverage is greatly biased in FAO-based cropland data due to false cropland expansion signals. The major reason is because of the distinct surveying methods used in China historically, as well as the political issues involved. For example, the amount of FAO-based (e.g. HYDE, LUH2) cropland abnormally increased by 28–32 Mha from 1980 to 1990, which contradicted the 4 Mha decline in cropland acreage revealed in our reconstructed cropland data in China (Yu et al. 2021). This is because the FAO data were reported from the Chinese Agricultural Yearbook, in which cropland underestimations have now been officially

acknowledged (Figure 2). More details about the biased sources can be found in Yu et al. (2021, 2022).

It should be point out that the cropland data is the basis for allocating N fertilizer use spatially. Due to such large biases, the existing, global N fertilizer products, which heavily relies on FAO-based cropland products (e.g. HYDE, LUH2), would inevitably inherit these biases in depicting historical N fertilizer use in China. Therefore, the existing products of N fertilizer use is expected to be 1) diluted spatially (due to lower but more extensive cropland distribution maps, which was also discussed in Tian et al. (2022). Please check the discussion in Lines480-488 in Tian et al. (2022)); and 2) distorted temporally (by the biased cropland area dynamics at grid-cell level, see differences between Figure 1e-h and Figure 1n-q).

All in all, we admit that our data is not perfect (and there is no perfect data), but this is one of the most updated and advanced datasets at present in China. It has corrected some of the most serious and commonly seen biases in existing products in China. We believe it could greatly improve the biogeochemical cycle-related simulations (e.g. $N_2O$ accounting in China), and we argue that the reconstructed data has the great value for future research.

References:

Tian, H., Bian, Z., Shi, H., Qin, X., Pan, N., Lu, C., ... & Zhang, B. (2022). History of anthropogenic Nitrogen inputs (HaNi) to the terrestrial biosphere: A 5-arcmin resolution annual dataset from 1860 to 2019. *Earth System Science Data Discussions*, 1-32.

Yu, Z., Jin, X., Miao, L., & Yang, X. (2021). A historical reconstruction of cropland in China from 1900 to 2016. *Earth System Science Data*, *13*(7), 3203-3218.

Yu, Z., Ciais, P., Piao, S., Houghton, R. A., Lu, C., Tian, H., ... & Zhou, G. (2022). Forest expansion dominates China's land carbon sink since 1980. *Nature Communications*, *13*(1), 1-12.

[Figure]

Figure 1. Comparisons of cropland coverage and changes during different periods in China. The left two columns are our reconstructed cropland maps, while the right two columns were derived from HYDE (FAO-based cropland maps). Image was obtained from Yu et al. (2021).

[Figure]

Figure 2. The accumulated changes of cropland areas in China (FAO-based cropland area is from LUH2-GCB, and it can be downloaded from https://luh.umd.edu/data.shtml; Yu's data is from Yu et al. 2021)

All in all, I don't think this article has enough innovation or contribution in terms of data source and data production method for publishing in the ESSD. The final data product was only qualified in terms of the average cropland fertilization, but the crop-specific fertilizer data was very crude, which greatly weakened the use value of the data. To sum up, I suggest rejecting the manuscript.

Response: Here we would like to re-iterate the innovation and contribution of our data. Our N fertilizer maps corrected the presence of serious biases in existing data products in both spatial and temporal at gridcell level. These corrections will greatly benefit the modeling community for greenhouse gas emission accounting, crop production evaluation, water pollution assessment, and vegetation growth simulations. In our former study, we made similar improvements with specific focus on contiguous

US (Cao, Lu, Yu 2018). In that study, we corrected biases in FAO-based cropland maps in the US, and reconstructed N fertilizer use using state-level inventory data (similar to provincial-level data in China), such as crop rotation and cropland area data. The N fertilizer product has been widely used in different studies (citation number for Cao, Lu, Yu (2018) is 156 as indicated in Google Scholar, which is ~39 citations annually). A simple example of our data's application is to benefit the global model intercomparison projects in carbon and nitrogen cycle simulations (e.g., NMIP: the global $N_2O$ Model Intercomparison Project, MsTMIP, and TRENDY project). These projects were often driven by 0.5 degree N fertilizer use data, which was derived from FAO-based cropland data (e.g. HYDE, LUH2, FAO stats: https://www.fao.org/faostat/en/). As we elaborated before, FAO-based products underestimated cropland coverage in traditionally cultivated areas, but it overestimated cropland coverage in low cultivated areas in China. Therefore, the N fertilizer input might be underrated in intensively cultivated areas in China (which is also found in this study - please see Figure 8). We believe the simulations (e.g., $N_2O$ accountings) will be improved in China if the N use data is updated.

Therefore, we would kindly advise the reviewer to revisit our revised manuscript and to reconsider for the final recommendation.

References:

Liu, Z., Yang, P., Wu, W., & You, L. (2018). Spatiotemporal changes of cropping structure in China during 1980–2011. *Journal of Geographical Sciences*, *28*(11), 1659-1671.

Lu, C., & Tian, H. (2017). Global nitrogen and phosphorus fertilizer use for agriculture production in the past half century: shifted hot spots and nutrient imbalance. *Earth System Science Data*, *9*(1), 181-192.

**Reviewer 3:**

This work reconstructed a historical, annual N fertilizers use dataset in China at 5 km × 5 km resolution covering the period of 1952 to 2018 by integrating improved cropland maps. The dataset is useful for many purposes. Generally the paper is well-written, I have several concerns for authors to improve it before accepting for pubilication.

Response: We thank the reviewer for valuing our work! We have addressed all the concerns raised by the reviewer.

1) The Figure 1. Methodology flowchart of nitrogen (N) fertilizer map reconstruction is not detailed enough to illustrate the methods used. Need to be improved by inclsuign the major methods used.

Response: We thank the reviewer for the suggestion. We agree and revised the methodology flowchart to help clear the methods used. Specifically, we added the temporal coverage of the data, the details of the processes, and the methods implemented. Please check our new flowchart below:

[Figure]

Figure 1. Methodology flowchart of nitrogen (N) fertilizer map reconstruction

2) Are the spatial pattern of N fertilizers use in each of the major crop types planted in China available? Why not resported in the main text? Are the data are availbale?

Response: We thank the reviewer for this precious suggestion. We have calculated the N fertilizer use in each of the major crop types in China. The data has now been deposited at https://doi.org/10.6084/m9.figshare.21371469.v1. Also, we also added discussion about this information using the crop-specific N fertilizer use in 2018 as the example. The figure and descriptions has also been added to the main text (Please see revisions in Lines210-214):

"We also showed the N fertilizer use rates in each of the major crop type in 2018 (Figure 6). Generally, the N fertilizer use was much higher in double-crop area than in monocrop area (Figure 6k vs Figure 6a-j). Moreover, early rice was seldom cultivated as monocrop in China (Figure 6a), which instead, was often planted with other crops in a year (part of the areas in Figure 6k). For monocrop areas, the N fertilizer use was found highest in corn (Figure 6e), while lowest N uses were detected in early rice and late rice (Figure 6a&6c)."

[Figure]

Figure 6. Spatial distribution of nitrogen fertilizer use rate by crop species (panels a-h indicate the fertilizer use rate in early rice, mid-season rice, late rice, wheat, corn, soybean, oil seeds, cotton, vegetable, other crops, and double crops, respectively; the

value in the scale bar indicates the N fertilizers use rate per square meter of land)

3) It is good that the authors compare the datasets wiith previous datasets. It is ture that the newly constrcued datasets are different but may need evidence that this data is more robust.

Response: We thank the reviewer for the suggestion. We have added the comparisons of the N fertilizer input at provincial derived from different studies (see figure below and the Figure 9 in the revised manuscript). Generally, we found that our maps (red color) perform better than the existing data products (blue color) in depicting the nitrogen fertilizer use in China. We believe these comparisons help evidence the quality of our data. Please also check our revisions in Lines256-260.

[Figure]

Figure. Comparisons of the total nitrogen fertilizer input in each province in China derived from different nitrogen fertilizer data products (the x-axis indicate the nitrogen

fertilizer input obtained from the Chinese statistical yearbook; the black line indicate 1:1 line, and the red and blue lines indicate the linear regressions of the provincial nitrogen fertilizer input derive from this study and other studies; panels a-d show comparisons in the period/years of 1994-2001, 2010, 2015, and 2018, respectively; data of other studies were derived from the nitrogen fertilizer maps of Potter et al. (2010), Nishina et al. (2017), Houlton et al. (2019) , and Tian et al (2022), respectively)